# Development of Sustainable and Cost-Competitive Injection-Molded Pieces of Partially Bio-Based Polyethylene Terephthalate through the Valorization of Cotton Textile Waste

**DOI:** 10.3390/ijms20061378

**Published:** 2019-03-19

**Authors:** Sergi Montava-Jordà, Sergio Torres-Giner, Santiago Ferrandiz-Bou, Luis Quiles-Carrillo, Nestor Montanes

**Affiliations:** 1Technological Institute of Materials (ITM), Universitat Politècnica de València (UPV), Plaza Ferrándiz y Carbonell 1, 03801 Alcoy, Spain; sermonjo@mcm.upv.es (S.M.-J.); sferrand@mcm.upv.es (S.F.-B.); nesmonmu@upvnet.upv.es (N.M.); 2Novel Materials and Nanotechnology Group, Institute of Agrochemistry and Food Technology (IATA), Spanish National Research Council (CSIC), Calle Catedrático Agustín Escardino Benlloch 7, 46980 Paterna, Spain

**Keywords:** bio-PET, cotton fibers, food packaging, biorefinery system design, waste valorization

## Abstract

This study presents the valorization of cotton waste from the textile industry for the development of sustainable and cost-competitive biopolymer composites. The as-received linter of recycled cotton was first chopped to obtain short fibers, called recycled cotton fibers (RCFs), which were thereafter melt-compounded in a twin-screw extruder with partially bio-based polyethylene terephthalate (bio-PET) and shaped into pieces by injection molding. It was observed that the incorporation of RCF, in the 1–10 wt% range, successfully increased rigidity and hardness of bio-PET. However, particularly at the highest fiber contents, the ductility and toughness of the pieces were considerably impaired due to the poor interfacial adhesion of the fibers to the biopolyester matrix. Interestingly, RCF acted as an effective nucleating agent for the bio-PET crystallization and it also increased thermal resistance. In addition, the overall dimensional stability of the pieces was improved as a function of the fiber loading. Therefore, bio-PET pieces containing 3–5 wt% RCF presented very balanced properties in terms of mechanical strength, toughness, and thermal resistance. The resultant biopolymer composite pieces can be of interest in rigid food packaging and related applications, contributing positively to the optimization of the integrated biorefinery system design and also to the valorization of textile wastes.

## 1. Introduction

The future scarcity of oil sources and the current strong awareness of waste disposal issues in modern society are two of the main drivers behind the interest, at both academic and industrial levels, in the use of bioplastics in a variety of consumer products. In spite of this, bioplastics still represent less than 1% of the approximately 300 million tons of plastics produced annually where packaging accounts for nearly 45% [1]. However, biopolymers will play an important role in the bioeconomy and undoubtedly shape the future of the packaging industry [2]. In this context, the European Commission (EC) has recently published a Packaging and Packaging Waste Directive laying out a strategy for plastics in a Circular Economy, which includes a ban on single-use plastics by the end of 2020 [3]. The term biopolymer is extensively used in the polymer literature when referring to both “bio-based” and “biodegradable” polymers [4]. Bio-based polymers are referred to any kind of polymer that is produced from renewable resources, which includes both naturally occurring polymers and synthetic polymers produced by means of monomers obtained from biological sources. Biodegradable polymers include those polymers whose physical and chemical properties undergo deterioration and completely degrade when exposed to microorganisms. Articles fully made of biodegradable polymers can be also compostable according to the specifications of international standards, for instance EN 13432 and ASTM D6400.

Bio-based but non-biodegradable polymers represent a group of biopolymers composed of both commodities and engineering plastics such as bio-based high-density polyethylene (bio-HDPE) [5], bio-based polypropylene (bio-PP) [6,7], bio-based polyamides (bio-PAs) [8], bio-based polyethylene terephthalate (bio-PET) [9], and, more recently, polyethylene furanoate (PEF) [10]. They can be either partially or fully synthesized from bio-based building blocks and offer almost identical chemical structure and properties than their petrochemical counterparts. The new branch of these biopolymers reflects the concept of the so-called “biorefinery system design” by which they are carbon neutral with the subsequent positive effect on greenhouse emissions and global warming [11,12]. According to European Bioplastics [13], the world production for these “green polymers” currently represents around 57% of the total worldwide bioplastics, that is, 1.2 metric tons (Mt). Among them, bio-PET is the most produced bioplastic, accounting for 26% of the total production. In the food packaging industry this has been recently evidenced by the Coca-Cola’s PET Plantbottle^®^ project where ethylene glycol (EG) and terephthalic acid (TA) are both derived from plant-based sugars and agricultural residues [14]. Despite current bio-PET’s popularity, only one of the two precursors, that is, EG, is produced from biomass whereas the other precursor, that is, TA, still remains fossil-based due to technical constraints [15,16]. Moreover, bio-PET at present presents a “GreenPremium” price, of approximately 2.25 €/kg, which is nearly 30% more expensive than its homologue polyethylene terephthalate (PET) of fossil origin [17].

In this context, lignocellulosic biomass obtained from plants, including forest residues, can be incorporated into biopolymers to further improve their sustainability and reduce cost. These include lignocellulosic fillers such as those obtained from the stalk, leaves, seeds, fruits, cereal straws, grass, etc. [18]. In the last years, different studies dealing with composite materials based on a PET matrix with different natural fillers have been reported. These materials cover a broad range of fillers, either from animals, such as spider silk [19] and lamb wool [20], or from plants, such as kenaf [21], sisal [22], pineapple [23], coir [24], coconut [25], softwood [26], hardwood [27], paper [28], bamboo [29], bagasse [30], and rice [31,32]. In addition, these natural fillers are habitually cost-effective materials and, in most cases, they are obtained from agro-food wastes or by-products of several industries. It is also worthy to note that the particular use of natural fibers (NFs) offer lower density and are also less abrasive that synthetic fibers that have been used to reinforce PET, including aramid fiber (AF) [33], glass fiber (GF) [34,35] and, more recently, carbon fiber (CF) [36]. Nevertheless, plant-derived fillers habitually offer limited thermal stability, increased water absorption, and show low compatibility with most of the polymer matrices so that the use of different compatibilizers is habitually needed [37,38]. All these drawbacks have restricted the use of NFs in high-performance and engineering applications, but their use in other applications such as food packaging is increasing [39].

Cotton, by contrast, is the most widely produced NF, accounting for more than 25 Mt/year according to the Discover Natual Fiber Initiative (DNFI) [40]. Cotton fiber is widely used in the textile industry and, subsequently, its waste generation is nowadays huge, reaching 4 Mt [41]. Final disposal of textile waste is causing soil contamination, canal obstructions, and drainage systems. Therefore, the valorization of industrial cotton waste currently represents a great opportunity to develop sustainable materials. The resultant recycled cotton fiber (RCF) is a multiuse product that is biodegradable and it can be reused even after being disposed of. Nevertheless, RCFs are nowadays being scarcely used in the industry as building materials, insulation panels, automotive interior parts, etc. [42]. Cotton waste is sold widely in large quantities and the estimated cost of recycled cotton from garments and mixed wastes of the textile industry can be as low as 0.03 €/kg [43]. Thus, the incorporation of such a cost-effective filler can help to reduce significantly the current raw material cost of biopolymers and then break the economical barrier to enter the food packaging market. In addition, cotton plants produce one of the purest forms of cellulose that allows its processing at high temperatures, which is the case of PET [44]. In relation to the end-of-life options, the post-consumer PET/cotton feedstreams are potentially non-recyclable or economically unattractive by means of depolymerization technologies, that is, chemical recycling [45]. However, the resultant RCF-containing PET waste can still be reused after washing and grinding, the so-called mechanical recycling, or melt-mixed with virgin and recycled PET to develop new composites [46]. For instance, cotton-based polymer composites can be melted with additives and fillers and then reprocessed into different articles for mainly non-food contact uses such as fibers for the textile industry, automotive parts, and industrial strapping [47]. Furthermore, as similar to other recycled polyester materials, the discarded composites can also be directly incorporated into existing post-consumer PET recycling streams at low contents, typically below 10 wt%, with negligible effect on PET’s mechanical performance [48,49].

The aim of this research work is to develop, for the first time, environmentally friendly and cost-effective composite materials using a bio-PET matrix and RCF obtained from textile wastes. To this end, different weight percentages of RCF were melt-compounded with bio-PET in a twin-screw extruder and then shaped into pieces by injection molding. The pieces obtained were characterized in terms of their mechanical, thermal, and thermomechanical properties in order to evaluate its potential in rigid food packaging and other related applications.

## 2. Results and Discussion

### 2.1. Visual Aspect and Density of Bio-PET/RCF Composite Pieces

Figure 1 shows the visual aspect of the material used and prepared in this study. Figure 1a shows the as-received cotton linter, which presented a dark grey color with different tonalities. The color is typical of wastes from blue jeans (also called Denim) and other types of clothing [42]. The chopped fibers were imaged after by optical microscopy and a representative image is shown in Figure 1b. The histogram shown in Figure 1c shows that the cotton-based fibers presented a similar diameter, varying in the range 10–30 μm and with a mean value of ~20 μm. Their relatively low fiber diameter and then high aspect ratio is a positive property for mechanical enhancement in polymer composites [50]. However, it also contributed to the observed high porosity of the fiber mats and hence to a relatively low bulk density, which limited the fibers feeding during extrusion to contents of up to 10 wt%. In another way, in Figure 1d one can observe the appearance of the neat bio-PET and bio-PET/RCF composite pieces varying the cotton-based fibers loading. PET is a semi-crystalline polyester and it can be manufactured into different packaging articles by selecting the appropriate cooling conditions resulting in either a full amorphous material, and thus very transparent, or highly crystalline, and then opaque and more heat resistant [51]. The neat bio-PET piece presented a natural color but it was also opaque, confirming that the biopolymer developed certain crystallinity during cooling in the injection mold and subsequent annealing. The pieces became darker as the weight percentage of RCF increased, developing an intense grey-to-brown color. Color changes of the pieces were measured and reported in Table 1 by the *L*a*b** coordinates. It can be observed that the luminance of neat bio-PET was approximately 75 while it progressively decreased down to values around 26 for the bio-PET/RCF composite pieces containing 5–10 wt% RCF. With regard to *a*b**, there was not a remarkable change in their values, presenting in all cases typical grey coordinates.

In terms of density, one can observe that the bio-PET/RCF composite pieces presented slightly higher densities that the neat bio-PET piece. This increase can be ascribed to the higher density of the cotton-based fibers than the biopolyester. In particular, the density of the material progressively increased from 1.253 g·cm^−3^, for the neat bio-PET piece, to a value of 1.301 g·cm^−3^, for the bio-PET/RCF composite piece at 10 wt%. In any case, the density values of the composite pieces prepared here are considerably lower than, for instance, those based on GF [52]. This can positively allow production of lightweight packaging articles with the advantage of energy saving.

### 2.2. Mechanical Properties of Bio-PET/RCF Composite Pieces

Table 2 summarizes the main mechanical properties of the bio-PET/RCF composite pieces. One can observe that the tensile strength and the elongation at break remarkably decreased with the RCF content. This can be mainly ascribed to the high difference in polarity between the typical hydrophobic nature of bio-PET and the extremely hydrophilic cotton fiber-based cellulose (~82.7%) and hemicelluloses (~5.7%) [44] and then rich in hydroxyl (–OH) groups on its surface. This particular composition of cotton also suggests that the fibers are relatively soft.

The maximum tensile strength (*σ*_max_), tensile modulus (*E*), and elongation at break (*ε_b_*) values of the bio-PET/RCF composite pieces were obtained through the tensile tests. As similarly reported for other NFs such as bamboo [53], the decrease in the tensile strength can be ascribed to the poor interaction between the NF surface (highly hydrophilic) and the surrounding polymer matrix (highly hydrophobic). In addition, this lack (or poor) interfacial adhesion is responsible for the appearance of micro-cracks and the stress concentration phenomenon, both promoting fracture [54,55,56]. These phenomena became more intense with increasing the RCF content. The unfilled bio-PET piece was characterized by a tensile strength of 50.7 MPa and the addition of only 1 wt% RCF decreased its tensile strength to 48.1 MPa, which represents a decrease of about 5%. As can be seen in the table, the bio-PET/RCF composite piece containing 10 wt% RCF showed a remarkable decrease in tensile strength, down to half the initial value of the neat bio-PET piece. With regard to elongation at break, the bio-PET/RCF composite piece also presented significantly lower values. In fact, the neat bio-PET piece presented an elongation at break of 378.4% and this value was reduced to 8.1% by the incorporation of only 1 wt% RCF. A clear decreasing tendency in ductility was seen with increasing the RCF content, down to values of 2.8% for the composite piece containing the highest fiber content, that is, 10 wt% RCF. This observation is related to the fact that elongation at break is very sensitive to the material’s cohesion and, as mentioned above, a poor (or even absence of) interaction at the bio-PET–RCF interface resulted in low adhesion with the subsequent negative effect on the mechanical response. Positively, the tensile modulus increased with the RCF loading, which is directly related to the ductility decrease since the reduction in elongation at break was remarkably higher than that observed for the tensile strength. Therefore, the lower values of elongation led to higher tensile modulus values. In particular, the modulus increased from 777 MPa (neat bio-PET piece) up to 1124 MPa (bio-PET/RCF composite piece with 10 wt% RCF). Similar results were found in the literature for other composite systems based on NFs in absence of compatibilizers [28,47,57,58,59]. The addition of RCF also provided an increase in the Shore D hardness values. As suggested by the tensile results, the injection-molded pieces became more brittle but also stiffer with the incorporation of RCF. For this reason, it was possible to observe an increasing tendency in hardness from 67, for the neat bio-PET piece, up to above 76, for the composites containing 10 wt% RCF.

Toughness of the pieces was also estimated by means of the impact strength measured with the Charpy method. Impact strength is directly related to the mechanical resistant properties, that is, tensile strength, and mechanical ductile properties, that is, elongation at break. As indicated previously, addition of RCF yielded a remarkable decrease in both mechanical properties and, as a consequence, the impact strength also decreased. In fact, the initial impact strength of the unfilled bio-PET piece was close to 3 kJ·m^−2^ and this value was reduced to values close to 1 kJ·m^−2^ for the composite pieces filled with 1–5 wt% RCF. Moreover, addition of 10 wt% RCF led to a dramatic decrease in toughness, showing a value of 0.3 kJ·m^−2^. It is well known that the impact strength of fiber-based composites is highly dependent on the interfacial interaction between the embedded fillers and the surrounding polymer matrix [60]. Therefore, impact energy can be thus dissipated by fiber detachment, fiber and/or matrix fracture, and fiber pull-out. Fiber fracture dissipates less energy in comparison to fiber extraction. The first mechanism is common in composites with good interface interactions while the second usually appears in composites with a poor interface. So that, this observation suggests that the main failure mechanism was fiber pull-out, which can be ascribed to poor interface interactions between the embedded RCF and the surrounding bio-PET matrix. Although mechanical ductile properties of all bio-PET/RCF composites were remarkable lower than those of neat bio-PET, the composite pieces with 1–5 wt% RCF showed a reasonable balance between mechanical strength and toughness.

### 2.3. Morphology of Bio-PET/RCF Composite Pieces

Figure 2 gathers the field emission scanning electron microscopy (FESEM) images corresponding to fracture surfaces of the injection-molded pieces of the bio-PET/RCF composites after the impact tests. Figure 2a shows the surface of neat bio-PET. One can observe that bio-PET is a ductile polymer, presenting different micro-crack fronts formed during fracture by mechanical impact. No remarkable signs of plastic deformation were observed due to the fracture surface corresponding to impact-fractured notched samples that broke with a relatively low energy absorption. The effect of the RCF addition modified the fracture surface of the pieces, shown in Figure 2b–g. As the RCF loading increased, more voids (seen as dark areas) between the bio-PET matrix and the embedded fibers were readily visible. Moreover, in Figure 2f it can also be seen the fiber pull-out phenomenon during fracture described above in the mechanical analysis. These voids and pull-outs were produced as a result of the poor interaction between the extremely high hydrophilic cellulosic reinforcement and the hydrophobic matrix, which has been previously reported in several studies dealing with composites based on cotton fibers [61,62]. These surface morphologies are in total agreement with the above-reported mechanical properties.

### 2.4. Thermal Properties of Bio-PET/RCF Composite Pieces

Figure 3 shows a comparative plot of the differential scanning calorimetry (DSC) thermograms during the second heating of the bio-PET/RCF composite pieces at different RCF loadings. Table 3 gathers the main thermal parameters obtained from the DSC curves. The glass transition temperature (*T_g_*) was observed as a step in the baseline located between 75–85 °C, showing a mean value of 81 °C for the neat bio-PET sample. The cold crystallization process can be related to the exothermic peak located between 135 and 160 °C with a maximum peak, the so-called cold crystallization temperature (*T_cc_*), at 151 °C for the neat bio-PET sample. Finally, the endothermic peak comprised between 225 and 255 °C corresponds to the melting process of the total crystalline fraction in bio-PET. The melting temperature (*T_m_*) of neat bio-PET was 246 °C.

The incorporation of RCF into bio-PET slightly reduced the *T_g_* values by approximately 2 °C, indicating that the fiber addition favored chain mobility of the biopolyester. This effect may be ascribed to the presence of residual additives (e.g., sulfur, naphthol, soaps, enzymes, and dyes) in the cotton-based fibers [63] that could potentially plasticize the bio-PET matrix. The cold crystallization phenomenon was only detected in the neat bio-PET sample, indicating that all the composite samples crystallized during cooling. This similar behavior has been reported in other research works suggesting that cellulose exerts a nucleating effect and thus leads to an increase in the crystallization rate [64,65,66,67]. In fact, cellulose can contribute to a heterogeneous crystallization process by acting as external nuclei for crystallization that will have a direct effect on the crystallites nature [68]. The nucleating effect of RCF on bio-PET can be further observed by the increase in the T_m_ values, of about 3–4 °C, as well as in the endothermic melting enthalpy (ΔH_m_). As a result, the X_c_ values increased from 25.4% (neat bio-PET) up to 29% (composite filled with 5 wt% RCF). Additionally, whereas the melting process of the neat bio-PET sample occurred in a single melt peak, all the bio-PET/RCF composites showed two overlapped peaks during melting. This double-melting peak phenomenon can be ascribed to the formation of crystalline structures with dissimilar lamellae thicknesses or the presence of crystallite blocks with different degrees of perfection [69,70,71,72]. A similar influence on the crystallization process of PET was previously reported by Souza et al. [73] by blending with polypropylene (PP).

One of the main problems related to manufacturing NF-based PET composites is the relatively high T_m_ value of the polyester, located in the 240–260 °C range as shown during DSC analysis. This fact implies the use of a processing temperature that is high enough to start thermal degradation of lignocellulose. Indeed, lignin starts its degradation at about 250 °C and its corresponding degradation process is prolonged up to temperatures around 450 °C [74]. Moreover, cellulose degrades in the 300–400 °C range and full decomposition takes place from 450 °C [22,75,76]. To ascertain the thermal stability of the neat bio-PET piece and the bio-PET/RCF composite pieces, thermogravimetric analysis (TGA) was carried out up to 700 °C and the resultant plots are gathered in Figure 4. In particular, Figure 4a shows the TGA thermograms while Figure 4b depicts the first derivative of the curves (DTG). The most relevant parameters regarding the thermal degradation, that is, *T_onset_*, which represents the temperature necessary to start thermal degradation, *T_deg_,* which corresponds to the maximum degradation rate of the samples, and the residual mass, are summarized in Table 4.

Several previous studies based on PET/cotton fibers composites have revealed that PET degrades in a single step weight-loss process, while cotton fiber shows two main weight loss processes located at 365 and 433 °C [77,78] in a nitrogen atmosphere. Interestingly, one can observe a delay in the onset degradation temperature of up to approximately 25 °C for the bio-PET pieces containing 3–5 wt% RCF. This indicates that the presence of RCF positively contributed to improving the thermal stability of bio-PET, which is a positive result particularly taking into account that the thermal analysis was carried out in air, that is, a more aggressive atmosphere than nitrogen. One can also observe three thermal stages during thermal decomposition of the samples. The first one occurred at 300–375 °C, which is related to thermal degradation of hemicelluloses and the start of cellulose degradation [74]. This step overlapped with the degradation onset of the polyester. The second and strongest peak corresponds to the main thermal decomposition of bio-PET, with a maximum degradation rate at 448 °C. The decomposition mechanism of thermoplastic polyesters consists of an hererolytic scission via a six-membered ring intermediate, where the hydrogen from a β-carbon to the ester group is transferred to the ester carbonyl, followed by scission at the ester links, producing compounds with carboxylic and vinyl end groups [79]. It has been also reported that, at higher pyrolysis temperatures, radical (homolytic) degradation pathways may also occur [80]. The third degradation stage was located between 500–600 °C, being mainly related to char decomposition as well as degradation of the remaining lignin [63].

### 2.5. Thermomechanical Properties of Bio-PET/RCF Composite Pieces

Dynamical mechanical thermal analysis (DMTA) is a high versatility technique to obtain the thermomechanical response of cellulose-based polymer composites in dynamic conditions [81]. Figure 5 shows the DMTA behavior of the bio-PET/RCF composite pieces. Figure 5a represents the variation of the storage modulus (*G’*) as a function of temperature. It can be observed that bio-PET was characterized by two main thermomechanical transitions with the temperature increase. Up to 70 °C, the variation in *G’* was nearly negligible thus indicating that the polyester is in its glassy state since this temperature range does not affect the mechanical properties. Then, a remarkable decrease in G’ occurred in the temperature range comprised between 70 and 90 °C. This process particularly involved a decrease of almost three orders of magnitude, which is representative for the glass-to-rubber transition of bio-PET. After this, an increase in *G’* was observed in the temperature range from 110 to 130 °C, which is attributable to the cold crystallization process of the biopolyester since crystals formation increases stiffness. With regard to the bio-PET/RCF composite pieces, one can observe that the *G’* values were significantly higher than that of the neat bio-PET piece in the whole temperature range. This behavior can be related to the higher crystallinity of the injection-molded pieces described above during the thermal analysis. Similar results have been reported by Marques et al. [82] who also showed an improvement of the storage moduli in poly(butylene adipate-*co*-terephthalate) (PBAT) by reinforcement with modified NFs. Another relevant finding is that cold crystallization nearly vanished or it was remarkably reduced in all the bio-PET/RCF composite pieces, which also agreed with the DSC results, since the pieces mainly developed crystallization during cooling due to the nucleating effect of the fibers.

Figure 5b shows the evolution of the damping factor (*tan δ*) as a function of temperature. *T_g_* can be related to the maximum peak of the *tan δ* curves, which corresponds to alpha (α)-transition of bio-PET. This value was ~81 °C for the neat bio-PET piece, which is identical to the *T_g_* value obtained by DSC. A slight decrease in *T_g_* of 2–3 °C was also observed in the case of the bio-PET/RCF composite pieces due to the above-reported secondary plasticizing effect of the cotton-based fibers. It is also worthy of note that the remarkable reduction of the α-peak for all the composite pieces, which indicates that the relaxation of the bio-PET chains was partially suppressed. This thermomechanical change further confirms the higher crystallinity achieved in the pieces containing RCF since a lower number of biopolymer molecules underwent α-transition in the amorphous region [83].

## 3. Materials and Methods

### 3.1. Materials

Bio-PET, commercial grade Bio-PET 001, was purchased at NaturePlast (Ifs, France). This resin has up to 30 wt% of natural origin and it is characterized by a density of 1.3–1.4 g·cm^−3^, an intrinsic viscosity between 75–79 mL·g^−1^, and a water content of less than 0.4%. According to the manufacturer, this grade is fully recyclable in the flow of petroleum-based PET.

Multicolored Linter Cotton Jeans was provided by Alcocertex S.L.U. (Alcocer de Planes, Spain). The product was supplied in the form of a recycled cotton linter composed of long yarns that was obtained as a waste from the industrial production of jeans. According to the manufacturer, the composition is 80% cotton and 20% mixture of acrylic and synthetic fibers. The cotton linter was chopped manually to obtain fibers, the so-called RCFs, with a mean length varying between 15 and 30 mm. The fibers present a true density between 1.5–1.6 g·cm^−3^ and a bulk density of 0.2–0.3 g·cm^−3^, as determined by ISO 1183 and ISO 60, respectively.

### 3.2. Manufacturing of Composite Pieces

Both bio-PET and RCF were first dried at 60 °C for 72 h in a dehumidifying dryer MDEO from Industrial Marsé (Barcelona, Spain). RCF of was incorporated up to 10 wt% into bio-PET by melt compounding in a twin-screw extruder from Construcciones Mecánicas Dupra, S.L. (Alicante, Spain) equipped with a screw diameter of 25 mm and a length-to-diameter (L/D) ratio of 24. All materials were fed through the main hopper, being previously pre-homogenized in a zipper bag. The selected rotating speed was set to 25 rpm and the temperature profile, from the feeding to the die, was: 240–245–250–255 °C. The extruded material was cooled in air conditions and then subjected to a second extrusion at the same conditions to improve the final quality, that is, the fiber dispersion. After this, the materials were cooled down in air and pelletized. Table 5 summarizes the labelling and compositions of the materials prepared.

The compounded materials were thereafter dried at 60 °C for 72 h due to the high hydrophilicity of cotton and the high sensitiveness of PET to hydrolysis. After this, standard pieces for characterization were obtained by injection molding in a Meteor 270/75 from Mateu & Solé (Barcelona, Spain). The temperature profile in the injection molding machine was set, from the feeding to the nozzle, to 240–245–250–255 °C. The resultant pieces were finally annealed at 60 °C for 72 h to further develop crystallinity, improve their dimensional stability, and remove any residual moisture.

### 3.3. Microscopy

Morphology of RCF was analyzed in a stereomicroscope system SZX7 model from Olympus (Tokyo, Japan) using an ocular magnifying glass of 50×. This was equipped with a KL1500-LCD light source. Fracture surfaces of the pieces after the impact tests were observed by FESEM in a CARL ZEISS Ultra-55 FESEM microscope from Oxford Instruments (Abingdon, UK). To provide conducting properties, the samples were previously covered with a with a 5–7 nm gold-palladium layer in vacuum conditions in a cathodic sputter-coater Emitech SC7620 from Quorum Technologies LTD (East Sussex, UK). The acceleration voltage during FESEM analysis was set to 2.0 kV.

### 3.4. Color Measurements

Changes in color were measured in a colorimetric spectrophotometer ColorFlex from Hunterlab (Reston, VA, USA). The selected color space was the chromatic model *L*a*b** or CIELab (spherical colour space). *L** stands for the luminance: *L** = 0 represents dark and *L** = 100 indicates clarity or lightness. The *a*b** pair represents the chromaticity coordinates: *a** > 0 goes to red, *a** < 0 goes to green; *b** > 0 goes to yellow, *b** < 0 goes to blue. Five different measurements were carried out on the injection-molded samples and the average color coordinates were calculated.

### 3.5. Density Measurements

True density was determined on injection-molded samples sizing 80 × 10 × 4 mm^3^. Measurements were performed in triplicate using a scale AG245 from Mettler-Toledo, Inc. (Schwarzenbach, Switzerland) following ISO 1183.

### 3.6. Mechanical Tests

Tensile properties of the injection-molded pieces were obtained in a universal test machine ELIB-50 from S.A.E. Ibertest (Madrid, Spain) following ISO 527-1:2012. The selected cross-head speed was 5 mm·min^−1^ and the load cell was 5 kN. Shore D hardness of the pieces obtained was measured in a 676-D durometer from Instruments J. Bot S.A. (Barcelona, Spain) as indicated in ISO 868:2003. The impact strength was obtained in a 1-J Charpy’s pendulum from Metrotec (San Sebastián, Spain) on notched pieces (“V” type notch with a radius of 0.25 mm), according to ISO 179-1:2010. All mechanical tests were carried out at room temperature, that is, 25 °C, and at least 6 samples of each material were tested.

### 3.7. Thermal Tests

Thermal characterization was carried out by DSC on a DSC821 from Mettler-Toledo, Inc. (Schwarzenbach, Switzerland). Small samples sizing 6–7 mg were subjected to a dynamic thermal program of three stages: an initial heating from 30 up to 280 °C, a cooling process down to 0 °C, and a second heating cycle from 0 up to 350 °C. The heating and cooling rates were set at 10 °C·min^−1^ and the atmosphere was nitrogen at a flow-rate of 66 mL·min^−1^. DSC runs were done in triplicate to obtain reliable results. The main thermal parameters were obtained from the second heating runs to remove the thermal history of the pieces. In addition, the bio-PET crystallinity was also calculated by the expression:(1)$$\%\chi_{c}=\frac{(\Delta H_{m}-\Delta H_{cc)}}{\Delta H_{100\%}\;\cdot W_{p}}\cdot 100$$
where:
%χ*_c_* = Degree of crystallinity (%)*W_p_* = Weight fraction of bio-PET (%)Δ*H_m_* = Melting enthalpy (J·g^−1^)Δ*H_cc_* = Cold crystallization enthalpy (J·g^−1^)Δ*H*_100%_ = Melting enthalpy of 100% crystalline PET = 140 J·g^−1^ [84,85]

Thermal degradation was evaluated by TGA in a TGA/SDTA851 thermobalance from Mettler-Toledo, Inc. Samples with an average weight of 4–5 mg were subjected to a temperature sweep from 30 up to 700 °C at a constant heating rate of 20 °C·min^−1^ in air atmosphere (50 mL·min^−1^). TGA runs were performed in triplicate.

### 3.8. Thermomechanical Tests

DMTA was carried out in a magnetic bearing rheometer AR-G2 from TA Instruments (New Castle, USA) with a special clamp system to work with solid samples in a combination of torsion and shear. Rectangular injection-molded pieces sizing 40 × 10 × 4 mm^3^ were subjected to a temperature sweep between 30 and 200 °C at a constant heating rate of 2 °C·min^−1^. The maximum shear deformation (%γ) was set to 0.1% and the experiments were carried out at a constant frequency of 1 Hz. DMTA runs were carried out in triplicate.

## 4. Conclusions

This work describes the development of composite pieces made of bio-PET filled with RCF obtained from wastes of the textile industry. The incorporation of RCF into bio-PET resulted in pieces with an intense grey-to-brown color and slightly higher densities. The composite pieces showed improved elasticity and hardness. However, the fibers also reduced tensile strength and also, to a higher extent, elongation at break and impact strength. Morphological analysis confirmed that the mechanical impairment attained was related to the poor biopolymer–fiber interactions that originated from the high hydrophilicity of RCF and the hydrophobic nature of bio-PET. Thermal analysis indicated that the fibers acted, even at low contents, as a nucleating agent for the biopolymer, increasing both the melting peak and percentage of crystallinity. Moreover, the thermal stability of bio-PET was improved by RCF, the pieces being thermally stable up to more than 350 °C. The thermomechanical properties fully agreed with the mechanical and thermal analyses and showed that the dimensional stability of the pieces was favored by the fiber loading.

Therefore, bio-PET pieces containing 3–5 wt% RCF showed a positive balance in terms of enhanced rigidity and thermal resistance. The materials developed herein can represent a sustainable and cost-effective solution of interest to the rigid packaging industry, which currently demands for high-performing biopolymer-based materials with a lower carbon footprint and at competitive prices. Illustrative examples may include rigid articles that do not require transparency such as bottles, canisters, boxes, cans, aerosol containers, and jars as well as disposable cups and cutlery. Nevertheless, future studies are needed to increase the fiber content in the composites and also to address their loss of mechanical strength and ductility. Furthermore, analysis of the barrier properties and specific migration tests should also be performed according to the targeted application.

## Figures and Tables

**Figure 1 ijms-20-01378-f001:**
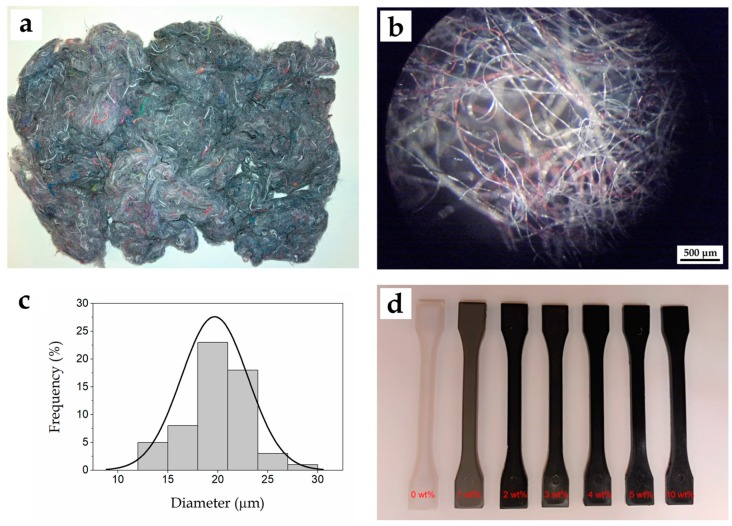
(**a**) As-received linter of recycled cotton; (**b**) optical microscopy image of recycled cotton fibers (RCFs) taken at 50× with scale marker of 500 μm; (**c**) fiber diameter histogram of RCF; (**d**) injection-molded pieces of bio-based polyethylene terephthalate (bio-PET) at different RCF contents.

**Figure 2 ijms-20-01378-f002:**
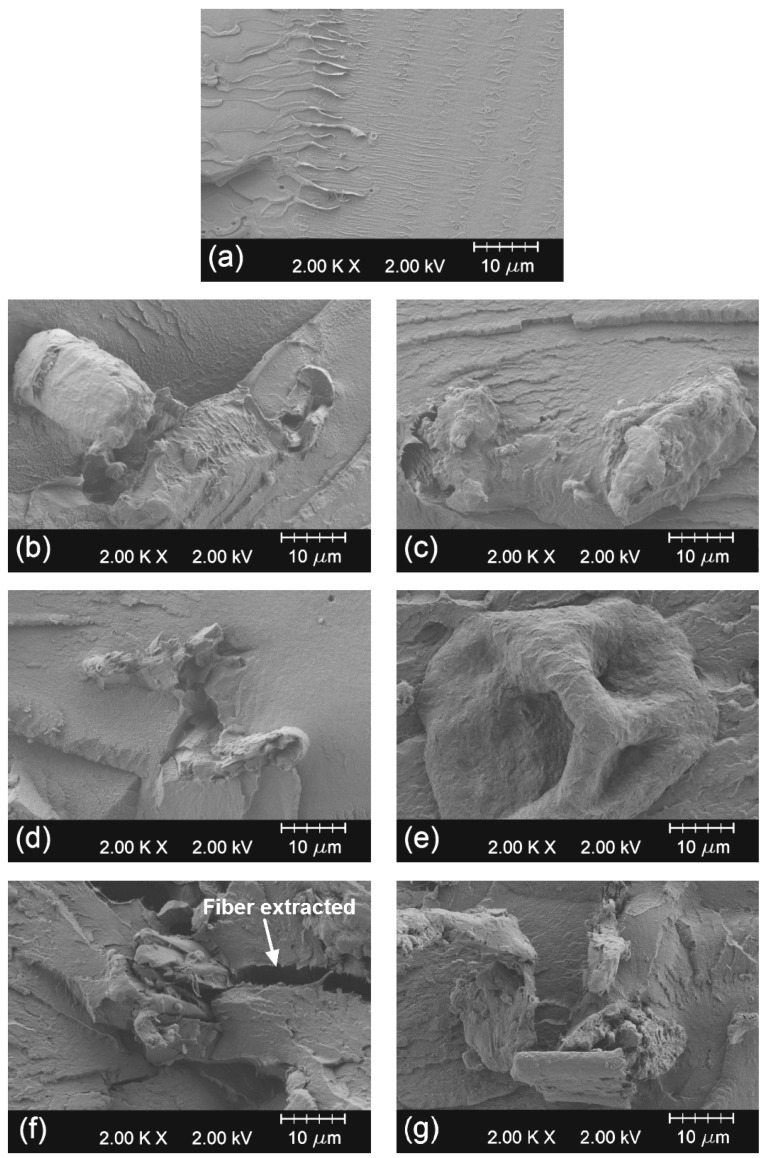
Field emission scanning electron microscopy (FESEM) images corresponding to the surface fractures of the bio-based polyethylene terephthalate (bio-PET)/recycled cotton fiber (RCF) composite pieces of: (**a**) Bio-PET100; (**b**) Bio-PET99/RCF01; (**c**) Bio-PET98/RCF02; (**d**) Bio-PET97/RCF03; (**e**) Bio-PET96-RCF04; (**f**) Bio-PET95/RCF05; (**g**) Bio-PET90/RCF10. Images were taken at 2000× and scale markers are 10 µm.

**Figure 3 ijms-20-01378-f003:**
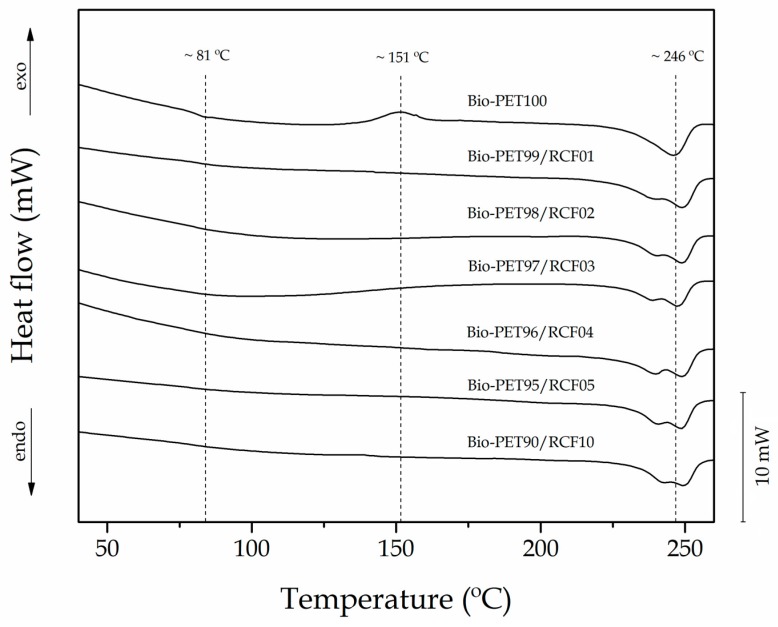
Differential scanning calorimetry (DSC) curves during second heating of the bio-based polyethylene terephthalate (bio-PET)/recycled cotton fiber (RCF) composite pieces.

**Figure 4 ijms-20-01378-f004:**
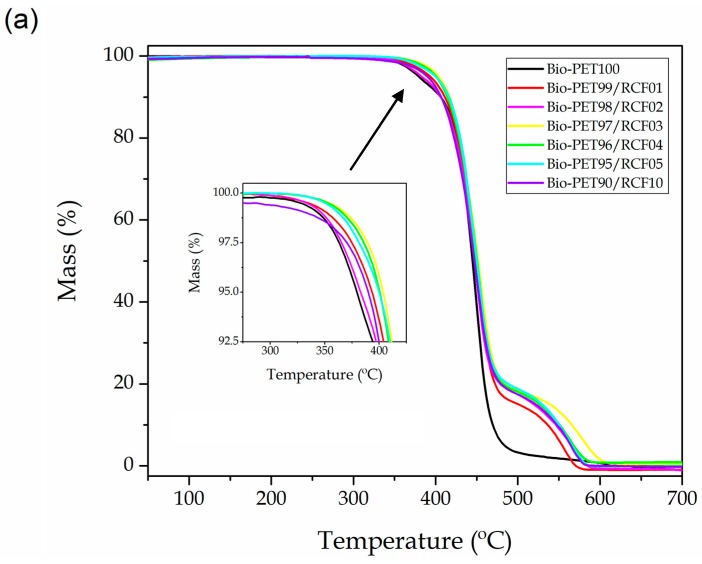
(**a**) Thermogravimetric analysis (TGA) and (**b**) first derivative thermogravimetric (DTG) curves of the bio-based polyethylene terephthalate (bio-PET)/recycled cotton fiber (RCF) composite pieces.

**Figure 5 ijms-20-01378-f005:**
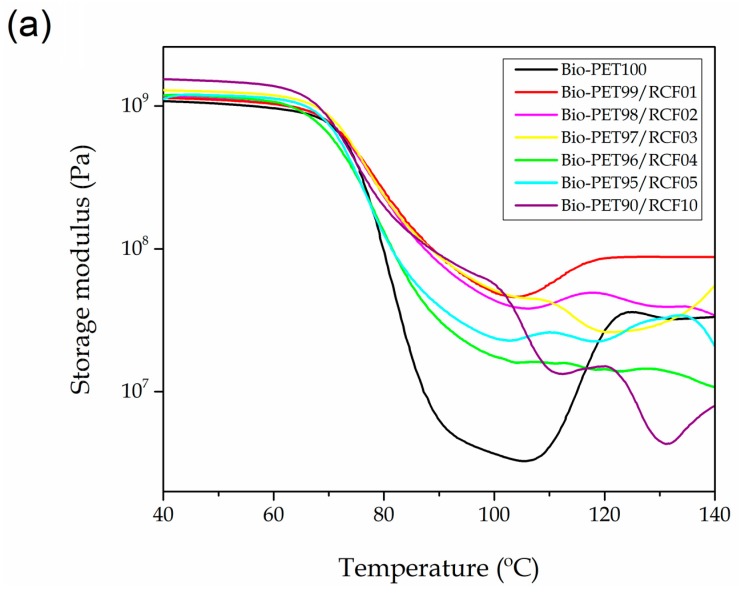
(**a**) Storage modulus (G′) and (**b**) damping factor (tan *δ*) of the bio-based polyethylene terephthalate (bio-PET)/recycled cotton fiber (RCF) composite pieces.

**Table 1 ijms-20-01378-t001:** Color coordinates by CIElab color space (*L***a***b**) and density of the bio-based polyethylene terephthalate (bio-PET)/recycled cotton fiber (RCF) composite pieces.

Code	*L**	*a**	*b**	Tone	Density (g·cm^−3^)
Bio-PET100	75.4 ± 1.0	−2.3 ± 0.3	−2.9 ± 0.4		1.253 ± 0.003
Bio-PET99/RCF01	46.8 ± 1.5	−1.2 ± 0.3	4.7 ± 1.8	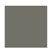	1.280 ± 0.002
Bio-PET98/RCF02	35.2 ± 1.0	−0.9 ± 0.1	5.2 ± 1.0		1.285 ± 0.011
Bio-PET97/RCF03	32.2 ± 0.8	−0.3 ± 0.2	5.0 ± 1.3		1.291 ± 0.004
Bio-PET96/RCF04	28.1 ± 1.6	−0.4 ± 0.1	3.4 ± 0.9		1.294 ± 0.001
Bio-PET95/RCF05	26.6 ± 1.3	0.5 ±0.1	1.6 ± 0.7		1.296 ± 0.002
Bio-PET90/RCF10	26.6 ± 2.5	0.3 ± 0.2	1.0 ± 0.4		1.301 ± 0.003

*L**: luminosity (+L luminous, −L dark); *a**: red/green coordinates (+a red, −a green); *b**: yellow/blue coordinates (+b yellow, −b blue).

**Table 2 ijms-20-01378-t002:** Summary of the mechanical properties of the bio-based polyethylene terephthalate (bio-PET)/recycled cotton fiber (RCF) composite pieces.

Piece	Tensile Strength (MPa)	Tensile Modulus (MPa)	Elongation at Break (%)	Shore D Hardness	Impact Strength (kJ·m^−2^)
Bio-PET100	50.7 ± 2.1	777 ± 58	378.4 ± 12.7	67.0 ± 2.7	2.97 ± 0.4
Bio-PET99/RCF01	48.1 ± 8.6	843 ± 83	8.1 ± 2.1	70.1 ± 0.2	1.06 ± 0.6
Bio-PET98/RCF02	42.9 ± 4.9	898 ± 102	6.5 ± 0.9	71.5 ± 0.6	0.96 ± 0.6
Bio-PET97/RCF03	39.7 ± 7.2	907 ± 82	6.2 ± 1.4	73.0 ± 0.7	0.96 ± 0.6
Bio-PET96/RCF04	36.7 ± 0.9	908 ± 18	5.7 ± 0.9	74.6 ± 0.5	0.91 ± 0.5
Bio-PET95/RCF05	29.9 ± 3.7	950 ± 44	4.2 ± 0.2	75.5 ± 1.0	0.91 ± 0.3
Bio-PET90/RCF10	24.4 ± 2.4	1124 ± 45	2.8 ± 0.8	76.3 ± 0.4	0.30 ± 0.1

**Table 3 ijms-20-01378-t003:** Summary of the main thermal properties of the bio-based polyethylene terephthalate (bio-PET)/recycled cotton fiber (RCF) composite pieces in terms of: glass transition temperature (T_g_), cold crystallization temperature (T_cc_), melting temperature (T_m_), cold crystallization enthalpy (ΔH_cc_), melting enthalpy (ΔH_m_), and degree of crystallinity (X_c_).

Piece	T_g_ (°C)	T_cc_ (°C)	T_m_ (°C)	ΔH_cc_ (J·g^−1^)	ΔH_m_ (J·g^−1^)	X_c_ (%)
Bio-PET100	81.3 ± 0.9	155.6 ± 0.9	245.6 ± 1.4	9.4 ± 0.1	26.2 ± 1.9	25.4 ± 1.4
Bio-PET99/RCF01	81.4 ± 0.4	-	248.6 ± 0.8	-	35.6 ± 2.5	25.7 ± 1.8
Bio-PET98/RCF02	80.7 ± 0.6	-	248.4 ± 0.4	-	36.5 ± 2.0	26.6 ± 1.5
Bio-PET97/RCF03	79.0 ± 2.7	-	247.1 ± 0.5	-	38.4 ± 1.9	28.2 ± 1.4
Bio-PET96/RCF04	80.1 ± 1.8	-	248.5 ± 0.3	-	38.8 ± 1.4	28.8 ± 1.1
Bio-PET95/RCF05	79.2 ± 0.2	-	248.1 ± 0.6	-	38.5 ± 0.3	29.0 ± 0.2
Bio-PET90/RCF10	79.4 ± 0.1	-	249.2 ± 0.4	-	35.0 ± 0.1	27.8 ± 0.1

**Table 4 ijms-20-01378-t004:** Summary of the main thermal properties of the bio-based polyethylene terephthalate (bio-PET)/recycled cotton fiber (RCF) composite pieces in terms of the: onset temperature of degradation (T_onset_), degradation temperature (T_deg_), and residual mass at 700 °C.

Code	*T_onset_* (°C)	*T_deg1_* (°C)	*T_deg2_* (°C)	*T_deg3_* (°C)	Residual Weight (%)
Bio-PET100	336.5 ± 1.3	-	448.0 ± 0.1	-	0.19 ± 0.03
Bio-PET99/RCF01	348.9 ± 1.9	380.0 ± 1.3	447.7 ± 0.1	555.0 ± 1.3	0.21 ± 0.01
Bio-PET98/RCF02	345.0 ± 3.6	375.3 ± 2.2	447.8 ± 0.1	564.3 ± 1.6	0.25 ± 0.02
Bio-PET97/RCF03	361.6 ± 2.2	389.3 ± 1.3	450.0 ± 1.3	576.0 ± 1.9	0.36 ± 0.04
Bio-PET96/RCF04	361.3 ± 1.6	373.0 ± 1.9	448.6 ± 1.6	562.0 ± 1.3	0.92 ± 0.03
Bio-PET95/RCF05	359.8 ± 1.9	370.7 ± 1.3	448.7 ± 0.1	562.0 ± 1.6	1.02 ± 0.02
Bio-PET90/RCF10	355.5 ± 1.3	380.0 ± 1.6	448.8 ± 0.1	587.7 ± 1.3	1.41 ± 0.05

**Table 5 ijms-20-01378-t005:** Codification and composition of the samples according to the content of bio-based polyethylene terephthalate (bio-PET) and recycled cotton fiber (RCF).

Sample	Bio-PET (wt%)	RCF (wt%)
Bio-PET100	100	0
Bio-PET99/RCF01	99	1
Bio-PET98/RCF02	98	2
Bio-PET97/RCF03	97	3
Bio-PET96/RCF04	96	4
Bio-PET95/RCF05	95	5
Bio-PET90/RCF10	90	10

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
