# Peer review of "Development of Sustainable and Cost-Competitive Injection-Molded Pieces of Partially Bio-Based Polyethylene Terephthalate through the Valorization of Cotton Textile Waste"

_ijms, 2019, doi:10.3390/ijms20061378_

Round 1
Reviewer 1 Report
The research is well presented, the introduction is extensive (almost a review), some spelling English errors are present.
The addition of cotton fibres in Bio-PET is claiming sustainability but is not addressing end of life. PET is recyclable the composite with natural fibres will not be, so which end of life? Incineration? Landfill? These would not be sustainable.
Table 1. Mechanical properties: is tensile Modulus elastic modulus? Three specimen are a bit few for an average (standard would be five), anyway Tensile strength decreases with increasing the fibre content, as normal with low matrix-fbre adhesion, but how can you say that you have reinforcing in conclusions? You say there that tensile strenght increases, this is not in table 1!!
Which cost saving do you expect if adding the fibres you increase density? Which is the cost of Bio-PET and the cost of the cotton fibres?
Do you have results with compatibilizers?
The paper is full fo characterization and results in crystallinity are interesting, but the high loss in mechanical propertis coupled with loss of recyclability of PET hardly justify for which reasons you should produce these composites= This need to be better addressed and the paper to be re focused in order to be published.
Author Response
The research is well presented, the introduction is extensive (almost a review), some spelling English errors are present.
The Introduction has been shortened and also refocused on the main topics and issues related to the research. English errors have been corrected.
The addition of cotton fibres in Bio-PET is claiming sustainability but is not addressing end of life. PET is recyclable the composite with natural fibres will not be, so which end of life? Incineration? Landfill? These would not be sustainable.
It is correct that post-consumer PET/cotton feedstreams are potentially no recyclable or economically unattractive by means of depolymerization technologies (chemical recycling). However, they can still be reused after washing and grinding (mechanical recycling) or melt-mixed with virgin and recycled PET to develop new composites. This statement is based on previous research studies addressing the recycling of PET composites and it has been included in the Introduction.
Table 1. Mechanical properties: is tensile Modulus elastic modulus? Three specimen are a bit few for an average (standard would be five), anyway Tensile strength decreases with increasing the fibre content, as normal with low matrix-fbre adhesion, but how can you say that you have reinforcing in conclusions? You say there that tensile strenght increases, this is not in table 1!!
Elastic modulus and tensile modulus is the same. The tensile tests were indeed performed on, at least, six samples, which has been amended in the experimental part (section 3.5). Whereas the tensile modulus and hardness increased after the incorporation of recycled cotton, tensile strength decreased. We have therefore modified the abstract and conclusions accordingly.
Which cost saving do you expect if adding the fibres you increase density? Which is the cost of Bio-PET and the cost of the cotton fibres?
There is direct cost saving resulted from the replacement of the biopolyester, which is currently around 30% more expensive than its petrochemical counterpart, with recycled cotton. This information has been included in the Introduction.
Do you have results with compatibilizers?
Based on the results of this study, we have recently started a new research activities focused on testing different reactive compatibilizers, trying to enhance the interfacial adhesion of the composites. We will also address the use of recycled polyesters. This information has been added in the Conclusion section.
The paper is full fo characterization and results in crystallinity are interesting, but the high loss in mechanical propertis coupled with loss of recyclability of PET hardly justify for which reasons you should produce these composites= This need to be better addressed and the paper to be re focused in order to be published.
The organization of the study has been restructured to show the benefits, that is, price reduction and valorization of an industrial waste. The issues of recyclability have also been discussed to support the development of the composites.
Reviewer 2 Report
The main important merit of the present paper consists of the very broad and detailed characterisation of complex plastic materials with the good motivation of using at least in part bioderived and recycled materials as the startin feed.
The main weak point is that the materials produced with very low content of recovered raw materials do not provide sufficent performance to justify such work from applied viewpoint even if providing some useful scientific information about properties correlation.
The paper should be ten exztensively revised and focused on scientific aspects .Some points are indicated to help this revision :
The introduction is to broad and long ; the information reported is largely available in manuìy papers and reviews and shopuld be limited to the real objective .
The starting material , commercial products, are not characterised adequately in term of composition and real biocontent. The presence of additives and other impurities would be also necessarely detremined
The t hermomechanical characterisation clearly demonstrates that the two materials are not compatible and no inrefacial interaction seems to occurr; then the mixing does not provide a composite but a filled PET.
The acceptable amount of used filler is very small and then the process does not look very useful for recycling.
During the mixing reactions between PET and OH goups of cotton may occurr . This would affect the final properties .Thus a detailed characteisation of the products by solvent extraction and spectrocopic analysis of the fractions collecyted would help understanding of the process aand porperties.
The use of a polymneric comptaibilizer ( diepoxxide or similar) would help and should be tested
Variatiobns of proertie swith composition is very modest and not regular. Reconsideration of the conclusion is recommended
Author Response
The introduction is to broad and long ; the information reported is largely available in manuìy papers and reviews and shopuld be limited to the real objective .
The Introduction has been shortened and limited to the main topics and issues of the research.
The starting material , commercial products, are not characterised adequately in term of composition and real biocontent. The presence of additives and other impurities would be also necessarely detremined
Full details of the bio-based content and properties of bio-PET is included in the experimental part. For the cotton waste, we have added all the available information provided by the manufacturer and also performed density measurements and optical microscopy. This new information is included in the Experimental Part, section 3.1 and 3.2, and in the Results, please see new section 2.1. These results have allowed to understand the effect of the fillers on both processability and properties.
The t hermomechanical characterisation clearly demonstrates that the two materials are not compatible and no inrefacial interaction seems to occurr; then the mixing does not provide a composite but a filled PET.
The manuscript has been modified indicating that the role of the cotton fibers was to fill the matrix without inducing any reinforcement.
The acceptable amount of used filler is very small and then the process does not look very useful for recycling.
The amount of incorporated fillers was maximized since the bulk density of the cotton waste was extremely low. Mechanical recycling is feasible and, therefore, the cotton-based composites can be melted with additives and fillers and then reprocessed into different articles for mainly non-food contact applications.
During the mixing reactions between PET and OH goups of cotton may occurr . This would affect the final properties .Thus a detailed characteisation of the products by solvent extraction and spectrocopic analysis of the fractions collecyted would help understanding of the process aand porperties.
The results showed that the fillers present a low interaction with the biopolymer matrix.Our group has a long-track expertise on the field and to achive these chemical interactions the use of reactive additives is usually needed.
The use of a polymneric comptaibilizer ( diepoxxide or similar) would help and should be tested
A new study on the use of reactive compatibilizers to improve the performance of these composites has been recently started as a continuation and extension of the present work. This information has been included ,as future work, in the Conclusion section.
Variatiobns of proertie swith composition is very modest and not regular. Reconsideration of the conclusion is recommended
The conclusion section has been modified accordingly.
Reviewer 3 Report
The strategy developed by the authors consists in blending recycled cotton waste (RCW) fibers with PET matrix. Such a strategy is not as original as the authors claim (page 5, line 1-5) and the addition of natural fibers as reinforcing agent to PET matrix is also a widely studied field. In addition, the authors also claim that these composites can be used on food packaging (page 1, topic; page 16, line 33-36). However, the gas barrier performance of these composites is unknown to reader in this study. While some details are still uncertain, I do think this manuscript is not adapted to International Journal of Molecular Sciences; therefore, I suggest the authors to consider another journal more oriented towards polymer composites.
Author Response
The strategy developed by the authors consists in blending recycled cotton waste (RCW) fibers with PET matrix. Such a strategy is not as original as the authors claim (page 5, line 1-5) and the addition of natural fibers as reinforcing agent to PET matrix is also a widely studied field. In addition, the authors also claim that these composites can be used on food packaging (page 1, topic; page 16, line 33-36). However, the gas barrier performance of these composites is unknown to reader in this study. While some details are still uncertain, I do think this manuscript is not adapted to International Journal of Molecular Sciences; therefore, I suggest the authors to consider another journal more oriented towards polymer composites.
It is correct that the polymer literature shows different composite materials based on PET and natural fillers (which are summarized in the Introduction). However, to the best of our knowledge, the use of recycled cotton from textile wastes in PET has not been studied. Moreover, the intended use of the composites is more oriented to rigid packaging applications, such as thick-wall food trays or containers, in which barrier properties are not as critical as those required in, for instance, films. This information has been included in the Conclusion section and the title has been modified to indicate the main achievements of the study.
Round 2
Reviewer 1 Report
The paper is signifcantly improved and fosued in the present form and is now suitable for publishing.
Reviewer 2 Report
I understand that the authors have extensively considered my remarks and have given sufficient response. The paper is now more valuable with adequate criticism to the still standing weak pointa about the value of the results.
Th paper can be published now even if a still ahve some doubts about the future useful application and would suggest to the authors to reduce the little too enthusiatic conclusion section.
Reviewer 3 Report
Although the authors have given some explanations, the manuscript has slightly changed with providing some additional sentences. Thus, I recommend this manuscript for publication without further changes.